# Evaluation of Two Different Approaches for Selecting Patients for Postoperative Radiotherapy in Deep-Seated High-Grade Soft Tissue Sarcomas in the Extremities and Trunk Wall

**DOI:** 10.3390/cancers16193423

**Published:** 2024-10-09

**Authors:** Andrea Thorn, Afrim Iljazi, Bodil Elisabeth Engelmann, Ninna Aggerholm-Pedersen, Thomas Baad-Hansen, Michael Mørk Petersen

**Affiliations:** 1Department of Orthopedic Surgery, Rigshospitalet, University of Copenhagen, Blegdamsvej 9, 2100 Copenhagen, Denmark; afrim.iljazi.04@regionh.dk (A.I.); michael.moerk.petersen@regionh.dk (M.M.P.); 2Department of Oncology, Herlev Hospital, University of Copenhagen, Borgmester Ib Juuls Vej 1, 2730 Herlev, Denmark; bodil.elisabeth.engelmann@regionh.dk; 3Department of Oncology, Aarhus University Hospital, Palle Juul-Jensen Blvd, 8200 Aarhus, Denmark; ninnpede@rm.dk; 4Department of Orthopedic Surgery, Tumor Section, Aarhus University Hospital, Palle Juul-Jensen Blvd, 8200 Aarhus, Denmark; thombaad@rm.dk

**Keywords:** sarcoma, high grade, local recurrence, survival, radiotherapy

## Abstract

**Simple Summary:**

This study compares two different strategies for using postoperative radiotherapy in patients with deep-seated grade 2–3 soft tissue sarcomas of the limbs and trunk wall. One treatment center routinely used postoperative radiotherapy for almost all patients (83%). The other center had a more restrictive approach to selecting patients for radiotherapy, and if the surgical margin was wider than 1 cm and/or a solid barrier was present, radiotherapy was not administered. This resulted in fewer patients (64%) receiving primary radiotherapy after surgery. By analyzing treatment outcomes over several years, we found no differences in local recurrence rates between the two approaches.

**Abstract:**

Two national sarcoma centers have had different approaches for selecting patients with grade 2–3 deep-seated soft tissue sarcomas (STS) for postoperative radiotherapy (RT). We evaluated potential differences in local recurrence in patients treated at the two centers. At Sarcoma Center 1 (SC1), RT was the standard treatment for all tumors except certain small tumors excised with a margin wider than 1 cm. Sarcoma Center 2 (SC2) avoided RT for tumors regardless of tumor size if removed with a margin wider than 1 cm and/or a solid barrier. We included 386 patients (SC1/SC2 = 196/190) over 18 years of age diagnosed with a non-metastatic grade 2–3, deep-seated STS of the extremities or trunk wall, who underwent primary surgical treatment (only tumors excised with a negative margin) from 1 January 2000, to 31 December 2016. Kaplan–Meier survival analysis, competing risk analysis, and cause-specific Cox regression were applied. A total of 284 patients received primary RT, 163 (83%) at SC1 and 121 (64%) at SC2 (*p* < 0.001). The cumulative incidence of local recurrence at five years was 15% (95% CI: 10–19%) at SC1 and 14% (95% CI: 9–19%) at SC2. Multivariate analysis showed no significant differences in local recurrence between the centers. We concluded that when entering all available patients into the analysis using an intention-to-treat principle, a more selective approach to postoperative RT in patients with grade 2–3 deep-seated STS did not lead to a higher local recurrence rate. However, with this study design, we cannot rule out if the local recurrence rate could have been lower if RT was administered to all tumors removed with a margin wider than 1 cm and/or a solid barrier.

## 1. Introduction

Sarcomas are a heterogeneous group of rare malignant tumors of mesenchymal cell origin in the musculoskeletal system, comprising 1% of all adult cancers [1]. Treatment of sarcomas requires a multidisciplinary, highly specialized team that evaluates the risks and benefits of all available options and expected outcomes to minimize the disease’s recurrence and preserve function and quality of life. The main treatment principle of soft tissue sarcomas (STS) of the extremities and the trunk wall, which has largely been unchanged since the 1980s, is surgery, often supplemented with radiotherapy (RT) [2].

Surgery combined with RT increases the probability of obtaining local control compared with surgery alone when treating deep-seated, grade 2–3 STS [3]. But even when using RT with 3D and intensity-modulated technique, it can lead to various complications in the treated area, such as wound complications, edema, joint stiffness, and rarely bone fractures [4,5].

In Denmark, sarcoma patients are mainly treated in one of only two national sarcoma centers. Until 2018, the two sarcoma centers had a different approach to selecting patients with deep-seated grade 2–3 STS treated with limb-sparing surgery with a negative margin for postoperative RT. In 2018, the Danish Sarcoma Group approved a new guideline dealing with RT of localized STS [6]. This guideline concluded that RT is the standard treatment in connection with limb-sparing surgery in non-metastatic deep-seated grade 2–3 STS, and the only exception for not using RT is in selected tumors, especially if below 5 cm in diameter and removed with a margin of more than 1 cm. One of the two sarcoma centers had followed an identical local guideline for many years before the new 2018 guideline was approved, while the other center had a more restrictive approach to the use of RT, thus often avoiding RT also in larger tumors if the surgical margin was considered sufficient.

We aimed to evaluate if a difference in local recurrence in patients with deep-seated grade 2–3 STS of the extremities and trunk wall could be detected between the two sarcoma centers with a known different policy regarding the use of RT.

Our primary hypothesis was that the local recurrence rate after limb-sparing surgical removal of deep-seated high-malignant STS of the extremities and trunk wall was lower at the center using RT in almost all patients.

## 2. Materials and Methods

### 2.1. Sarcoma Centers

This is a nationwide retrospective cohort study based on data from both national sarcoma centers in Denmark. The two sarcoma centers employed different approaches in selecting patients with localized deep-seated grade 2–3 STS for postoperative RT after surgical removal of tumors with a margin classified in the pathology report to be negative:

Sarcoma Center 1 (SC1): RT was the treatment of choice for all patients, and the only exception where it could be considered not to use postoperative RT was in small tumors (<5 cm in diameter) removed with a margin wider than 1 cm.

Sarcoma Center 2 (SC2): RT was usually used for all patients, but not in tumors, regardless of tumor size, removed with a margin wider than 1 cm or if a significant barrier, e.g., fascia or periosteum/bone, was a part of the margin.

Both centers used the Musculoskeletal Tumor Society (MSTS) system for margin classification [7]. Because of the well-known difficulties in determining the surgical margin [8], we included patients for this study if their tumors were removed with a negative margin (marginal or wide based on information from the pathologist reports). Thus, the surgeries included corresponded well to an R0 resection, according to the American Joint Committee on Cancer (AJCC) staging system [9].

### 2.2. Identifications for Study Cohorts and Study Endpoints

Since 2009, all Danish bone sarcoma and STS patients have been reported prospectively to The Danish Sarcoma Registry (DSR). DSR contains information about patient characteristics, tumor characteristics, diagnostics, details on treatment, local and distant recurrence, comorbidity, and death [10]. Before 2009, patients treated at SC1 were reported prospectively to The Aarhus Sarcoma Registry (ASR), a locally based registry validated in 2013 with a completeness of 99.3% [11]. Data from patients treated at SC2 from 2000 to 2009 were retrospectively collected based on information from the local pathology register. Patient information was collected from patient records, the radiotherapy system Aria, and The Danish CPR Registry [12]. We selected the following study endpoints for comparison between the two sarcoma centers, SC1 and SC2:

Primary endpoint:5-year risk of local recurrence.

Secondary endpoints:Local recurrence rate for the entire observation period.Proportion of patients that received postoperative RT.Five-year overall survival.Overall survival for the entire observation period.

### 2.3. Patients

A total of 2337 patients diagnosed with STS of the extremities and trunk wall from 2000 to 2016 in Denmark were identified from DSR, ASR, and the local pathology register. We defined the following anatomic tumor locations of our study population:Upper extremity tumor: from or distal to the shoulder.Lower extremity tumor: from the pelvic area to the toes (excluding genital and peritoneal tumors).Trunk wall tumors: from the clavicle to the top of the pelvis (excluding tumors located in the mamma, retroperitoneal, intraabdominal, head, and neck).

We established our study population using the following exclusion criteria:4.Patients who did not undergo surgical treatment of their STS.5.Grade 1 or borderline tumors (Trojani grading system [13])6.Tumors that were not considered deep-seated. We defined deep-seated tumor location as having a tumor located under or through the facia.7.Patients operated on in another hospital than SC1 or SC2.8.Patients younger than 18 years at the time of operation.9.Patients with a tumor removed with an intralesional margin (defined by the pathologist).10.Patients that had received pre- or postoperative chemotherapy within three months of the primary operation.11.Patients that had received preoperative radiation.12.Metastases within three months of surgery.13.Patients who had an amputation as primary surgery.14.Patients operated on for a local recurrence.

The exclusions left us with a study population of 386 patients with deep-seated, grade 2 + 3 STS of the extremities or trunk wall operated on with a negative margin in Denmark from 2000 to 2016, 196 at SC1 and 190 at SC2 (Figure 1).

Time to local recurrence was measured from the time of operation at one of the sarcoma centers until the date of the pathology report that confirmed local recurrence.

Overall survival time was measured from the surgery date until death from any cause, emigration, or end of follow-up time (1 January 2023).

### 2.4. Statistical Analysis

Patient baseline characteristics are presented with descriptive statistics. Continuous variables are presented as either mean and range or median and range, while categorical variables are presented as the total number and percentage of the total. The difference between groups was tested using the Wilcoxon rank sum test for continuous variables and the χ^2^ test for categorical variables. We applied an intention-to-treat-like principle in the analysis of our primary outcome, thus including all patients treated in the 2 centers regardless of whether RT was received or if the planned dose was delivered. We calculated the local recurrence rates using Competing Risk Analysis with Aalen–Johansen’s estimator, where death was treated as a competing risk. At the same time, the difference between centers was tested using Gray’s test. We estimated the overall survival (OS) probability with the Kaplan–Meier estimator and tested for a difference between the two centers with the log-rank test. We employed Cause-Specific Cox Regression to investigate the potential association between the treatment approach, characterized by stratification for the sarcoma center, and the association with local recurrence. We performed a crude analysis only stratifying for the treatment center and a multivariate analysis adjusting for well-known prognostic covariates. The multivariate model was constructed by including the following predefined variables that are known to influence local recurrence: age, gender, location, surgical margin, tumor size, and histological grade. We assessed the proportional hazard assumption with the complementary log–log regression and found no violations. The results of the regression analyses were presented as the hazard ratio (HR) with 95% confidence intervals (95-CI). We considered a *p*-value < 0.05 as statistically significant. Analyses were performed using R, version 4.2.0 (R Development Core Team, Vienna, Austria, 2020).

### 2.5. Ethics

The study was approved by the Danish Data Protection Agency (Videnscenter for Dataanmeldelser) (P-2022-92) and by the Center for Regional Udvikling, sundhedsforskning og innovation (R-22008540).

## 3. Results

### 3.1. Patient and Tumor Characteristics

Table 1 shows the characteristics of the patients who met the inclusion criteria. The mean age for the overall population was 61 (18–95) years; 54% were males, 86% of tumors were 5 cm or bigger, and 71% were in the lower extremity.

Statistically significant differences between the patients treated with postoperative radiation at the two centers were found regarding histological grade, surgical margin, and tumor location. SC2 had a larger number of histological grade 2 tumors (50 (41%) versus 32 (19%)), and SC1 had more patients where the tumor was removed with a wide margin (90 (55%) versus 26 (21%)). Only a difference in age and grade could be found in patients not treated with post-operative radiation, with patients in SC1 being older (69 vs. 60 years) and SC1 having more grade 3 tumors (76% vs. 55%). (Table 1).

The five most common overall histological types were sarcoma without specifications NOS, malignant fibrous histiocytoma (MFH), leiomyosarcoma, myxoid liposarcoma, and undifferentiated pleomorphic sarcoma. The distribution was quite similar in the two treatment centers, with sarcoma NOS being the most common type at both SC1 (17%) and SC2 (30%) (Table 2).

### 3.2. Non-RT Patient Characteristics

At SC1, 33 patients did not receive primary radiotherapy after their surgery. In nine patients, an explanation could not be found in the patient record; one of these patients was later diagnosed with local recurrence. Two patients reportedly did not receive primary RT as the surgical margin was considered sufficient to skip radiation therapy, and none experienced local recurrence. Sixteen patients were excluded from primary RT due to clinical factors such as prolonged wound complication, old age with multiple comorbidities, or severe psychological disease. Three of these were later diagnosed with local recurrence. Four patients did not receive primary RT due to patient factors such as patient refusal or psychosocial comorbidity; two later were diagnosed with local recurrence. Two patients died before RT could be performed (Table 3).

At SC2, 69 patients did not receive primary radiotherapy following surgery. Three patients had no documented explanation for omitting primary radiotherapy; two experienced local recurrence. A total of 45 patients did not receive primary radiotherapy due to surgical margin considered sufficient to skip radiation therapy, and of these patients, seven (15.5%) developed local recurrence. Eleven patients did not receive primary radiotherapy due to clinical factors. Five of these patients developed local recurrence. Eight patients declined primary radiotherapy or had psychosocial factors precluding treatment, and of these three patients experienced local recurrence. Two patients died before radiotherapy could be performed (Table 3).

### 3.3. Local Recurrence, Amputation, and Overall Survival

Of the 386 patients in our study population, 67 (17%) developed a local recurrence, 36 (18%) patients at SC1, and 31 (16%) at SC2 (*p* = 0.6). We found the cumulative incidence of local recurrence at five years to be 15% (CI-95: 10–20%) at SC1 and 14% (CI-95 9–19%) at SC2, with no statistically significant difference between the two centers (*p* = 0.6) (Figure 2).

A total of 284 (74%) patients received primary postoperative RT, of which 163 (83%) patients were treated at SC1 and 121 (64%) at SC2 (*p* < 0.001).

Six patients were secondarily amputated due to local recurrence. Only two patients had not received radiation and were in the no-radiation group at SC2. Both patients were in the group that did not receive radiation because of patient-related factors, and they both received secondary radiotherapy before amputation was decided.

A total of 229 (59%) patients were dead at the end of follow-up, 123 (62%) at SC1 and 106 (56%) at SC2, with no significant difference between the two treatment centers (*p* = 0.2). The median length of follow-up for the survivors was 11 years (1–23). One patient was lost to follow-up after 1 year, and one person emigrated after 3.6 years. All other patients had a minimum follow-up time of 5 years or until death.

The Kaplan–Meier 5-year probability of OS for the entire cohort was 60% (CI-95: 55–65%). We stratified according to treatment centers and found that the 5-year survival rates were 58% (CI-95: 51–65%) and 63% (CI-95: 56–70%) for SC1 and SC2, respectively. A log-rank test comparing both centers for the entire study period showed no difference in OS (*p* = 0.2) (Figure 3).

### 3.4. Cause-Specific Cox Regression

The crude cause-specific HR for local recurrence was 15% lower for SC2 patients compared to SC1 (HR: 0.85; CI-95: 0.52–1.37), and the adjusted HR for local recurrence was 17% lower for SC2 compared to SC1 (HR: 0.83; CI-95: 0.50–1.38). However, both failed to meet the threshold for statistical significance. Only tumors located in truncus compared to upper (HR: 0.40, CI-95: 0.18–0.87) and lower (HR: 0.34, CI-95: 0.18–0.64) extremities were found to be statistically significant prognostic factors of local recurrence (Table 4).

## 4. Discussion

We found a 5-year risk of local recurrence of 15% for SC1 and 14% for SC2. The 5-year survival rate was 58% for patients treated at SC1 and 63% for those treated at SC2. This is very similar to what is reported in other studies for survival (62–77%) and local recurrence when focusing on patients with grade 2 + 3 deep-seated STS [14,15,16,17,18,19]. However, in prospective studies where RT is administered to all patients, it has been possible to achieve local recurrence rates below 10% [20,21]

Despite a statistically significant difference in the utilization of postoperative RT between the two sarcoma centers, we did not find any difference in the local recurrence rates. After adjusting for multiple clinical factors like age, gender, location, surgical margin, tumor size, and histological grade, no significant difference in HR for local recurrence could still be found between centers, although these factors are well-known prognostic indicators for local recurrence in STS based on previous studies [22,23,24,25,26]. In the cause-specific Cox regression, only location (upper and lower extremity versus truncus) had a higher HR in connection with local recurrence.

RT is a well-documented modality for improving local control in STS of the extremities and trunk wall [27,28]. When comparing patient characteristics between the treatment centers for patients who received RT, we found no differences in age, sex, or tumor size. There was, however, a statistically significant difference in the number of patients receiving primary RT, 83% versus 64% at SC1 and SC2, respectively. In practice, patients do not always receive the treatment prescribed by their doctors, either due to clinical factors that deem the treatment impossible or due to patient compliance. Out of 33 patients who did not receive radiation at SC1, 14 were due to clinical factors, including postoperative wound complications, comorbidities, and age, and 4 were due to patient factors such as patient refusal. These factors and the fact that SC1 guidelines allowed for certain tumors <5 cm to potentially forgo radiation explain why the percentage of patients receiving radiotherapy does not reach 100% in SC1. In contrast, out of the 69 patients at SC2 who did not receive radiation, for 45 patients this was because the surgical margin was considered sufficient to skip radiation therapy, and 16% of them experienced local recurrence afterward. Only ten patients at SC2 did not receive radiation due to factors outside the control of the clinician.

Because of STS’s rarity, many studies on STS suffer from limitations such as small sample sizes and a lack of standardized patient selection criteria. These studies are prone to bias and encompass various histological grades, surgical margins, tumor locations, and depth. A notable strength of the present study is the inclusion of a large number of patients from a relatively modern cohort.

Although SC1 and SC2 follow the same criteria and guidelines for tumor classification and grading, the individual pathologist’s assessment can vary slightly depending on who conducts the evaluation, as well as the fact that certain tumors are only graded as high-malignant and not graded further. The WHO classification of STS published in the Autumn of 2002 [29], together with advances in immunohistochemistry, led to substantial changes in STS diagnosis, resulting in a significantly reduced amount of fibrosarcoma and malignant fibrous histiocytoma (MFH) [30] Thus, the difference between centers regarding the STS diagnoses can be that one center was slower to adapt to the new sarcoma classification. However, the treatment protocol for the various subtypes in deep-seated grade 2 + 3 tumors has not been different if it was a grade 2 or grade 3 tumor. Other studies also argue that local recurrence rates mainly depend on other clinical factors, such as size, and not the specific grade [31].

A registry study, such as ours, relies on precise data entry to uphold study reliability and quality. Our national sarcoma registry predominantly relies on data entered manually, posing a heightened risk of information bias. To mitigate this bias, we adopted a strategy where the same individual collected and verified data accuracy, and then an oncologist at each center verified the patients who did not receive RT. Through this approach, we ensured the quality and adequacy of data of this registry and minimized potential bias. Our study is retrospective, and no randomization was performed; therefore, a selection bias may have occurred. We have tried to moderate this with our multivariate analysis, including many potential prognostic covariates.

Information regarding patients’ comorbidities was unavailable and thus not included in this study. Comorbidities have shown diverse effects on predicting OS in STS [15,32,33]. It remains unclear whether the presence of comorbidities indirectly affected patient selection for RT. Metastatic disease at the time of diagnosis is a well-established negative prognostic marker for sarcoma patients [34]. Consequently, patients with metastatic disease were excluded regardless of whether they underwent oncological intervention.

We also encountered challenges with the oncological data, particularly for patients from the earlier part of the cohort, where specific RT-related details—such as total dose, beam energy, and fraction size—were missing for a substantial part of the patients. We were, therefore, unable to conduct analyses that considered these factors. As a result, we adopted the retrospective intention-to-treat-like principle to address this limitation for our analyses. This meant that patients were assessed based on the treatment group they were initially assigned to according to the center’s different selection criteria, regardless of whether they adhered to that treatment, changed their treatment during the study period, received different RT in the two centers or treatment protocols were changed over time. Because of this approach, our results primarily focus on the local recurrence outcomes at the two centers concerning whether patients were determined to receive RT rather than the specific details of their RT regimen.

It is important to acknowledge that STS are rare and represent various histological subtypes that, in turn, display a range of anatomical and pathological differences. Although our study focused on grade 2 + 3 and deep-seated tumors only, we included a diverse range of histological diagnoses. Due to the limited number of patients within each subtype, specific calculations on histological subtypes could not be performed. Some studies have found that specific subtypes have a higher risk of local recurrence and should potentially be treated more aggressively than other subtypes [35,36,37,38]. However, most studies have a small number of patients with various subtypes. Considering this, it is not surprising that the literature shows conflicting results regarding mechanisms that affect local recurrence.

Our multivariate analysis did not find evidence supporting surgical margin (marginal or wide) as a significant factor for local recurrence. However, this might not be surprising since we excluded tumors removed with an intralesional margin. Thus, the study was not designed to evaluate the influence of marginal status on local recurrence.

Our study highlights that surgery alone, with a wide surgical margin, may suffice without the need for radiation for a subgroup of patients. This notion is supported by other studies comparing groups of patients treated with surgery plus RT versus surgery alone, which found no significant difference in local recurrence rates [39,40,41,42]. However, it is crucial to consider this within the broader context of other clinical factors.

## 5. Conclusions

We found that a more restrictive approach for selecting patients with high-malignant deep-seated STS for postoperative RT, resulting in fewer patients (64% versus 83%) receiving postoperative RT, did not result in any statistically significant difference in local recurrence rate between the two sarcoma centers. However, with this current study design, we cannot rule out the possibility that the local recurrence rate could have been lower if RT was administered to all tumors removed with a margin wider than 1 cm and/or a solid barrier.

## Figures and Tables

**Figure 1 cancers-16-03423-f001:**
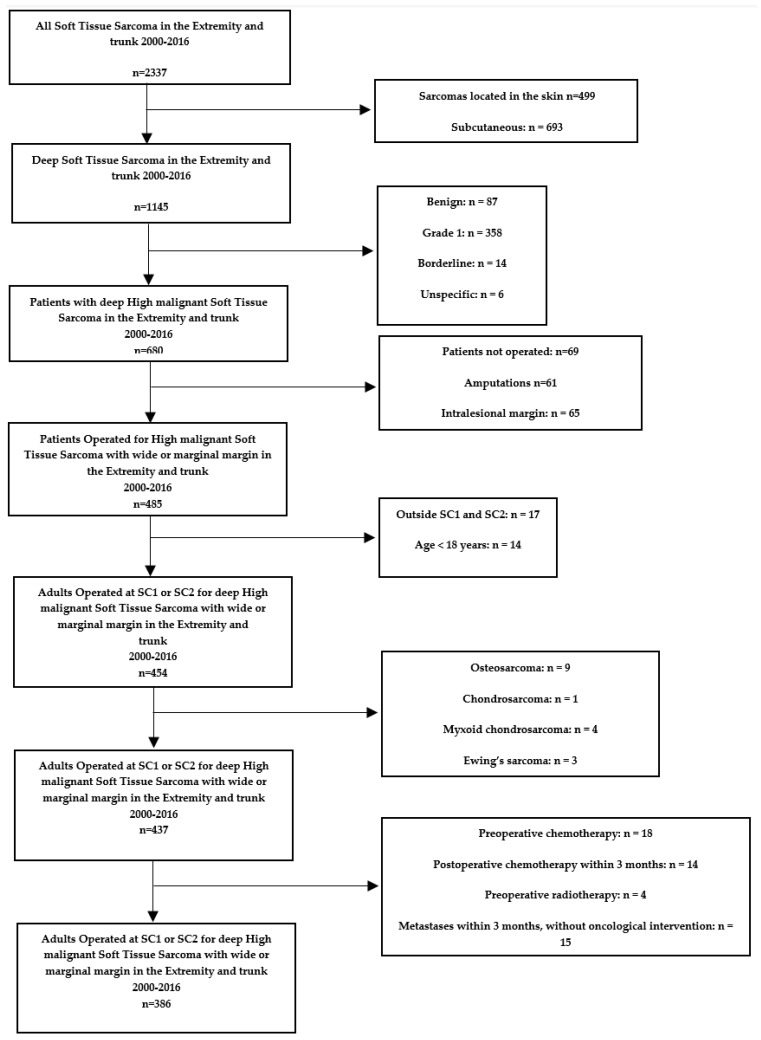
Exclusion flowchart.

**Figure 2 cancers-16-03423-f002:**
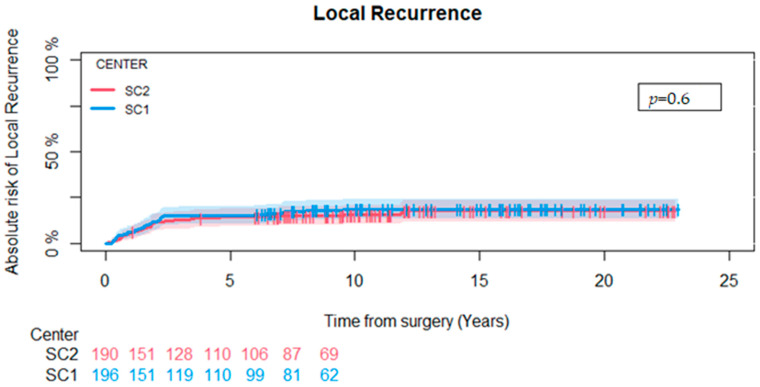
Competing risk analysis for local recurrence comparing SC1 and SC2. No difference could be found between the two centers (*p* = 0.6).

**Figure 3 cancers-16-03423-f003:**
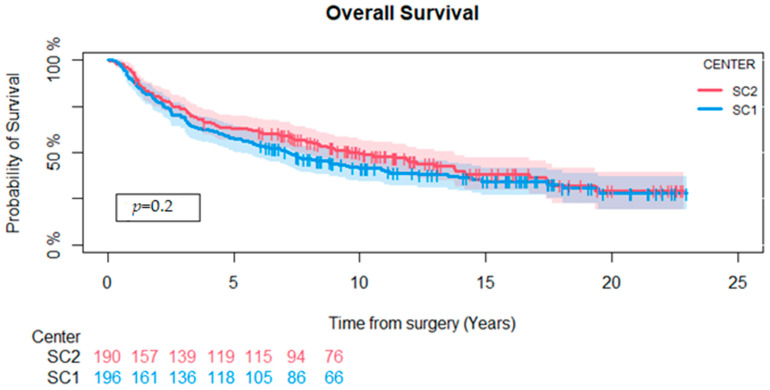
Kaplan–Meier survival analysis comparing overall survival at Sarcoma Centers 1 and 2. There was no difference between the two centers (*p* = 0.2).

**Table 1 cancers-16-03423-t001:** The table describes and compares demographics and clinical variables divided by radiation treatment and centers. The difference between centers was tested using the Wilcoxon rank sum test for continuous variables and the χ^2^ test for categorical variables.

Radiation		Yes			No		
Treatment Center	Overall	SC1	SC2	*p*- Value ^2^	SC1	SC2	*p*- Value ^2^
**Overall**	(*n* = 386)	(*n* = 163)	(*n* = 121)		(*n* = 33)	(*n* = 69)	
**Sex**				0.6			0.8
Female	176 (45%)	81 (50%)	56 (46%)		12 (36%)	27 (40%)	
Male	210 (55%)	82 (50%)	65 (54%)		21 (64%)	41 (60%)	
**Age (years) ^1^**	61 (18–95)	60 (19–86)	60 (18–85)	0.7	69 (27–92)	60 (18–95)	0.034
**Histological grade**				<0.001			0.044
Grade 2	121 (32%)	32 (20%)	50 (41%)		8 (24%)	31 (45%)	
Grade 3	265 (68%)	131 (80%)	71 (59%)		25 (76%)	38 (55%)	
**Location**				0.046			0.7
Lower Extremity	273 (71%)	105 (64%)	93 (77%)		24 (74%)	51 (74%)	
Truncal	42 (11%)	16 (10%)	11 (9%)		4 (12%)	11 (16%)	
Upper Extremity	71 (18%)	42 (26%)	17 (14%)		5 (15%)	7 (10%)	
**Tumor size**				0.054			0.087
<5 cm	55 (14%)	26 (16%)	10 (8%)		3 (9%)	16 (23%)	
≥5 cm	331 (86%)	137 (84%)	111 (92%)		30 (91%)	53 (77%)	
**Surgical margin**				<0.001			0.4
Marginal	204 (53%)	74 (45%)	96 (79%)		13 (39%)	21 (30%)	
Wide	182 (47%)	89 (55%)	25 (21%)		20 (61%)	48 (70%)	

^1^ Age: mean (range), frequency. ^2^ Pearson’s Chi-squared test; Wilcoxon rank sum test.

**Table 2 cancers-16-03423-t002:** The five most common histological subtypes and the distribution between treatment centers.

Treatment Center	Overall	SC1	SC2
**Overall**	(*n* = 386)	(*n* = 196)	(*n* = 190)
**Sarcoma NOS**	90 (23%)	33 (17%)	57 (30%)
**Malignant fibrous histiocytoma**	37 (10%)	23 (12%)	14 (7.5%)
**Leiomyosarcoma**	36 (9%)	24 (12%)	12 (6%)
**Myxoid liposarcoma**	34 (9%)	13 (6.5%)	21 (11%)
**Undifferentiated pleomorphic sarcoma**	29 (8%)	15 (7.5%)	14 (7.5%)
**Other**	160 (41%)	88 (45%)	72 (38%)

**Table 3 cancers-16-03423-t003:** Overview, clinical reasoning, and local recurrence of all patients not receiving primary postoperative radiotherapy. [Number of local recurrences].

Treatment Center	Overall *n* = 102	SC1 *n* = 33	SC2 *n* = 69
**Surgical margin considered sufficient to skip radiation therapy**	47 (46%) [7]	2 (6%) [0]	45 (65%) [7]
**Clinical factors** **(wound complications, co-morbidity)**	27 (26%) [6]	16 (48%) [2]	11 (16%) [5]
**Patient factors** **(patient refusal or psychosocial comorbidity)**	12 (12%) [5]	4 (12%) [2]	8 (12%) [3]
**Death before radiation**	4 (4%) [0]	2 (6%) [0]	2 (3%) [0]
**No explanation**	12(12%) [3]	9 (27%) [1]	3 (4%) [2]

**Table 4 cancers-16-03423-t004:** Cause-specific hazards regression analysis for local recurrence adjusting for age, gender, location, surgical margin, tumor size, and histological grade. Only tumor location (truncus) displayed statistically significant increases in the HR regarding local recurrence.

	Hazard Ratio; (95% CI)	*p* Value
**Crude**		
Center (reference = SC1)	0.85; (0.52–1.37)	0.5
**Adjusted**		
Center (reference = SC1)	0.83; (0.50–1.37)	0.48
Age at diagnosis	1.02; (1.00–1.03)	0.06
Tumor location (Truncus vs. lower extremity)	0.34; (0.18–0.64)	<0.001
Tumor location (Truncus vs. upper extremity)	0.40; (0.18–0.87)	0.02
Higher histological grade (grade 2 vs. grade 3)	1.07; (0.63–1.83)	0.76
Sex (male)	1.32; (0.80–2.16)	0.28
Size (>5 cm)	1.42; (0.64–3.15)	0.38
Surgical margin (wide vs. marginal)	0.96; (0.58–1.59)	0.86

## Data Availability

The data presented in this study are restricted due to patient confidentiality and are, therefore, not publicly available. However, they can be made available to the corresponding author upon reasonable request.

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
