# Peer review of "Evaluation of Two Different Approaches for Selecting Patients for Postoperative Radiotherapy in Deep-Seated High-Grade Soft Tissue Sarcomas in the Extremities and Trunk Wall"

_cancers, 2024, doi:10.3390/cancers16193423_

Round 1
Reviewer 1 Report
Comments and Suggestions for Authors
We applaud the authors sarcoma centers for maintaining an excellent quality Sarcoma Registry as a rich source of patient data. However, the fundamental issue with this manuscript is that authors propose that the overall similarity in outcome at 2 centers where the rate of radiation administration was 88% vs. 70% is sufficient to justify broadly forgoing a standard-of-care treatment approach that has pre-existing randomized data to support its utility. An alternative approach that would have been much more impactful was to specifically evaluate the subset of patients with clearly pre-defined criteria who meet the “borderline” situation for meriting radiation in these authors opinion at one institution and compare their outcomes to a matched cohort of patients at the other institution. While not definitive such an analysis would be much more thought provoking as delineating if radiation can be forgone and for which specific cohort it cane be forgone. As it stands, the majority of patients at both centers (with patients at both centers have significantly different patient and tumor characteristics) received radiation which dilutes the ability to say anything about the effectiveness of radiation.
Additional issues include:
- In the introduction the authors discuss the “controversial” effect of RT on overall survival. However, as outlined by Baldini et al in the reference cited, the rationale for RT is not typically improving survival but rather improving local control and limb preservation (which is associated with major quality of life and functional benefits). Framing the paper around the issue of whether RT improves survival really misses the point.
- It is interesting that the authors conducted this study only on the subset of patients who did not receive pre-operative radiation. The Danish Sarcoma Group guidelines as well as the more recent ASTRO guidelines outline the toxicity advantages of preoperative RT. It would be interesting to know why this subset of patients did not receive pre-op.
- The distribution of relevant risk factors between SC1 and SC2 appear quite different. With SC2 having significantly more lower grade tumors, younger patients and narrower margins, these 2 cohorts seem very different and hard to meaningfully compare in a retrospective manner, even with multivariate analysis.
- The authors note that for SC1, the only exception to RT being delivered “was in small tumors (<5 cm in diameter) removed with a margin wider than 1 cm” but in Table 1 it appears that 91% of patients in SC1 who did not receive RT had tumors >=5 cm which is the opposite of what the predefine “rules” for this institution would suggest.
- As discussed above, discussing overall survival as a major endpoint (it’s actually discussed before local recurrence in Section 3.2) doesn’t make sense. Typically overall survival is driven by distant metastatic disease which isn’t even looked at in this dataset.
- Rather, the authors should report long-term limb preservation rates in both institutions with both treatment approaches.
- Similarly it would be more interesting to lump all of SC1 and SC2 patients together and then evaluate local recurrence specifically in those who received vs. did not receive RT and then perform multivariate analysis to predict for local recurrence as the primary endpoint.
- It is odd that the authors describe local recurrence as their primary endpoint but Table 3 evaluates predictors of overall survival (which as stated above) would not be expected to be significantly effected by radiation. I would suggest removing this table entirely focusing on Table 4 and focusing on +/-RT across both centers as a variable of interest.
- In both cohorts, local recurrence rates seem higher than expected. In O’Sullivan et al, local recurrence appears <10% and while follow-up is limited in more recent studies of hypofractionated RT, local recurrence generally seems more similar to that rate. In fact the rates in this paper are similar to what would be suggested by the nomogram for patients not receiving RT produced by Cahlon et al in 2012.
Comments on the Quality of English LanguageThere are minor English verbiage issues that could easily be addressed (e.g. “high-malignant” should be grade 2-3, “only tumors excised with a marginal or wide-margin” should be only tumors excised with a negative margin)
Author Response
Dear Reviewer,
Thank you for your comments and for the opportunity to revise our manuscript. We greatly appreciate your constructive feedback and have carefully addressed your suggestions.
A detailed response to all your questions is provided in the attached PDF.
Sincerely,
Andrea Thorn

Reviewer 2 Report
Comments and Suggestions for Authors
Thank you for inviting me to evaluate the article, “Evaluation of two different approaches for selecting patients for postoperative radiotherapy in deep-seated high-grade soft tissue sarcomas in the extremities and trunk wall”. This study aimed to evaluate a difference in clinical outcome of patients with deep-seated high-grade soft tissue sarcoma of the extremities and trunk wall, in comparison between the two sarcoma centers with a different policy regarding the use of postoperative radiation.
This might be a study of interest to specialists in this area, but should not be suitable for publication in Cancers. I will give some suggestions.
Major comment
1. As authors discussed, this article does not contain detailed information of radiotherapy, including total dose, beam energy, and fraction size. Therefore, the data in this study cannot be conclusive. In addition, the indication of postoperative radiation in two centers are not significantly different. At the very least, authors should evaluate using cases with similar radiation dose/fraction in both centers, and present the data.
2. Authors should evaluate the data in comparison using histological subtypes. Myxoid liposarcoma is one of well-known radio-effective subtypes but MFH/UPS is not, and the subtype should be one of factors whether or not to employ postoperative radiation.
Minor comments
In Table 1, overall number of No-radiation in SC1 is 28, but the sum in each variable is 23. This should be amended.
Author Response

(The authors gave the same response as above.)

Round 2
Reviewer 1 Report
Comments and Suggestions for Authors
The authors have done an excellent job addressing the issues that were brought up. I think the one major missing component is adding to the discussion a frank assessment that LR rates here are higher than what has been reported in the prospective literature (with citations of that prospective literature as well as other institutional data). The authors can explain why they think this is but it is important to explicitly note that 14-15% LR appears to be significantly higher than we would expect. If a paragraph such as that were added I think it would be reasonable to consider this for publication.
Author Response
Dear Reviewer,
Thank you for your valuable feedback and for recognizing our efforts in addressing the issues raised. We appreciate your insightful suggestion to include a discussion on the local recurrence (LR) rates in our study, particularly the comparison to those reported in prospective literature.
In response, we have added a paragraph to the discussion acknowledging that the observed LR rates of 14-15% in our study are higher than expected. We have cited relevant prospective studies, including O’Sullivan et al. (2002), which demonstrate that LR rates below 10% can be achieved when radiotherapy is administered to all patients.
We hope this revision addresses your concerns. We are grateful for your thoughtful input, which has strengthened the manuscript.
Thank you again for your time and effort
Reviewer 2 Report
Comments and Suggestions for Authors
Thank you for inviting me to re-evaluate the article, “Evaluation of two different approaches for selecting patients for postoperative radiotherapy in deep-seated high-grade soft tissue sarcomas in the extremities and trunk wall”.
Comment:
The manuscript has been revised well. I believe this manuscript will be acceptable for publication in Cancers.
Author Response
Dear Reviewer,
Thank you for your positive feedback and for taking the time to re-evaluate our manuscript, “Evaluation of Two Different Approaches for Selecting Patients for Postoperative Radiotherapy in Deep-Seated High-Grade Soft Tissue Sarcomas in the Extremities and Trunk Wall.”
We are pleased that you find the revisions satisfactory and that the manuscript is now acceptable for publication. We appreciate your constructive comments throughout the review process, which have helped us improve the quality of the manuscript.
Thank you again for your time and effort.